# Somatic burden in Russia during the COVID-19 pandemic

**Alena Zolotareva**[1]*, **Anna Khegay**[1], **Elena Voevodina**[1], **Igor Kritsky**[2], **Roman Ibragimov**[2], **Nina Nizovskih**[3], **Vsevolod Konstantinov**[4], **Arina Malenova**[5], **Irina Belasheva**[6], **Natalia Khodyreva**[7], **Vladimir Preobrazhensky**[1], **Kristina Azanova**[1], **Lilia Sarapultseva**[8], **Almira Galimova**[9], **Inna Atamanova**[10], **Anastasia Kulik**[11], **Yulia Neyaskina**[11], **Maksim Lapshin**[12], **Marina Mamonova**[13], **Ruslan Kadyrov**[14], **Ekaterina Volkova**[14], **Viktoria Drachkova**[7], **Andrey Seryy**[15], **Natalia Kosheleva**[1], **Evgeny Osin**[1,16]

1 International Laboratory of Positive Psychology of Personality and Motivation, HSE University, Moscow, Russia, 2 Institute of Natural Sciences and Mathematics, Ural Federal University Named After the First President of Russia B.N. Yeltsin, Ekaterinburg, Russia, 3 Department of Psychology, Vyatka State University, Kirov, Russia, 4 Department of General Psychology, Penza State University, Penza, Russia, 5 Department of Psychology, Dostoevsky Omsk State University, Omsk, Russia, 6 Department of General Psychology and Personality Psychology, North-Caucasus Federal University, Stavropol, Russia, 7 Department of Psychology, Saint-Petersburg State University, Saint-Petersburg, Russia, 8 Department of Mathematics and Natural Sciences, Russian State Vocational Pedagogical University, Ekaterinburg, Russia, 9 Department of Theory and Technology of Social Work, Samara National Research University, Samara, Russia, 10 Department of Genetic and Clinical Psychology, Tomsk State University, Tomsk, Russia, 11 Department of Theoretical and Practical Psychology, Kamchatka State University Named after Vitus Bering, Petropavlovsk-Kamchatskiy, Russia, 12 Russian-Chinese Education and Research Center of System Pathology, South Ural State University, Chelyabinsk, Russia, 13 Research Laboratory for Integrative Traditional Chinese and Western Medicine, South Ural State University, Chelyabinsk, Russia, 14 Department of General Psychological Disciplines, Pacific State Medical University, Vladivostok, Russia, 15 Department of Psychology, Kemerovo State University, Kemerovo, Russia, 16 Laboratory LINP2, University of Paris Nanterre, Paris, France

* alena.a.zolotareva@gmail.com

**Data Availability Statement:** The study data are available at https://osf.io/9hw52/.

**Funding:** The study was supported by HSE University Basic Research Program.

## Abstract

Somatic burden has become one of the most common psychological reactions to the COVID-19 pandemic worldwide. This study examined the prevalence of somatic burden, latent profiles, and associated factors of somatic symptoms during the pandemic in a large sample of Russians. We used cross-sectional data from 10,205 Russians collected during October-December, 2021. Prevalence of somatic burden was assessed with the Somatic Symptom Scale-8. Latent profiles of somatic burden were identified using latent profile analysis. Multinomial logistic regression was used to examine demographic, socioeconomic, and psychological associated factors of somatic burden. Over one-third (37%) of the Russians reported being somatised. We selected the three-latent profile solution with high somatic burden profile (16%), medium somatic burden profile (37%), and low somatic burden profile (47%). The associated factors of greater somatic burden were female gender, lower education, history of COVID-19 disease, refusing vaccination against SARS-CoV-2 infection, poorer self-rated health, greater fear of COVID-19 pandemic, and living in regions with higher excess mortality. Overall, this study contributes to knowledge about the prevalence, latent profiles, and associated factors of somatic burden during the COVID-19

**Competing interests:** NO authors have competing interests.

pandemic. It can be useful to researchers in psychosomatic medicine and practitioners in the health care system.

## Introduction

The COVID-19 pandemic, with over 526 million confirmed cases and over six million deaths (World Health Organization data as of May 29, 2022), has caused serious damage to the mental and physical health of the world's population. Numerous studies suggested that 30.7% of persons experience somatic symptoms, which exceeds the general prevalence of some other mental health issues, for instance 16.4% for suicide ideation, 6.4% for obsessive-compulsive symptoms, 25.7% for panic disorder, 2.4% for phobia anxiety, 22.8% for adjustment disorder, and 1.2% for suicide attempts [1].

Many mental health issues can be expressed psychosomatically with the experience of somatic symptoms replacing or combining with the symptoms of psychological distress [2]. Somatic symptoms during the first waves of the COVID-19 pandemic were reported by 23.8% of Spaniards [3], 29% of Germans [4], 31.1% of Iranians [5], 45.9% of Chinese [6], and 62.6% of Brazilians [7]. Physical complaints were most frequently manifested in vulnerable groups, such as health-care workers [8], cancer patients [9], chronic disease patients [10], and patients awaiting transplantation [11]. Sometimes somatic symptoms were observed during the recovery period after the COVID-19 disease. Three months after hospitalization for OVID-19, 23.5% of patients reported body aches, 20.3% fatigue, 19% shortness of breath, and 13.1% headaches [12]. Interestingly, persistent physical complaints expressed 10 to 12 months after the first wave of COVID-19 pandemic were more strongly associated with the person's belief in having experienced COVID-19 disease than with having had a laboratory-confirmed SARS-CoV-2 infection [13].

Before the pandemic, some researchers studied somatic symptoms using latent class analysis (LCA) or latent profile analysis (LPA). These analyses are variants of a person-centered statistical approach aiming to find out subtypes of related cases in empirical data and establish whether there are subgroups of individuals with similar symptom profiles [14]. Generally, researchers derived between three and eight latent classes of somatic symptoms. In one study, a three-class solution revealed subgroups with low, moderate, and high psychosomatic burden differing mostly in the extent, rather than in the qualitative combinations of somatic symptoms [15]. In another study, a five-class solution detected subgroups without any health problems, with multiple somatic symptoms, and with specific symptom patterns reflecting abnormal tiredness, gastrointestinal problems, and pain-related symptoms [16]. In a third study, an eight-class solution identified subgroups without somatic symptoms, with a high burden of all somatic symptoms, three intermediate classes with specific symptoms of muscle and joint pain, gastrointestinal symptoms, general symptoms, and three intermediate classes characterized by combinations of specific somatic symptoms [17].

We aimed to replicate and extend these previous findings by examining the prevalence of somatic burden, investigating the latent profiles and predictors of somatic symptoms during the COVID-19 pandemic in a large sample of Russians. Specifically, we expected that somatic burden during the COVID-19 pandemic should be associated to some extent with known pre-pandemic risk factors and demographics, such as female gender, older age, not being in a partnership, lower education background [18–20], but might be more dependent on health- and pandemic-related risk factors, such as poorer self-rated health, history of COVID-19 disease, refusing vaccination against SARS-CoV-2 infection, greater fear of COVID-19 pandemic,

living in regions with higher excess mortality, and preventive behavior during the COVID-19 pandemic.

## Methods

### Participants

The online quantitative survey "Somatic Burden in Russia" recruited volunteers who were at least 18 years old and resident in Russia between October and December, 2021. Participants who did not meet these criteria were excluded from the survey. The survey included a participant information sheet explaining the study aims and objectives, an informed consent page, a sociodemographic form, and questionnaires measuring self-rated health, somatic burden, fear of COVID-19, and preventive behavior. The survey invitation was advertised using social media popular in Russia, including Instagram, Facebook, VKontakte, Telegram, and WhatsApp groups.

### Measures

The respondents completed a sociodemographic form followed by a one-item self-rated health measure aimed to evaluate the general health status as "poor", "fair", "good", "very good", or "excellent" [21]. Next, the survey included the following questionnaires:

Somatic Symptoms Scale (SSS-8) [22]. The SSS-8 assesses eight somatic complaints during the past seven days (stomach or bowel problems; back pain; pain in arms, legs, or joints; headaches; chest pain or shortness of breath; dizziness; feeling tired or having low energy; trouble sleeping). Symptoms are scored on a five-point response option from 0 ("not bothered at all") to 4 ("bothered very much"). The cut-off score for somatic burden is $\geq 12$, with the severity of somatic symptoms characterized as minimal (0–3 points), low (4–7 points), medium (8–11 points), high (12–15 points), or very high degree of somatic symptoms (16–32 points). In the present sample, the translated Russian version of the SSS-8 showed good reliability (Cronbach's $\alpha = 0.831$).

Fear of COVID-19 Scale (FCV-19S) [23]. The FCV-19S is a seven-item scale measuring psychological and physiological responses to fear associated with the SARS-CoV-2 infection and COVID-19 pandemic. Responses can range from 1 ("strongly disagree") to 5 ("strongly agree"). We used the Russian version of the FCV-19S [24]. In the present sample, the FCV-19S showed good reliability (Cronbach's $\alpha = 0.831$).

OVID-19 Preventive Behavior Index (CPBI) [25]. The CPBI is a ten-item scale evaluating preventive behaviors aimed at reducing exposure to the SARS-CoV-2 infection and COVID-19 pandemic (wearing a facemask, regular hand hygiene, maintaining social distance, avoiding any non-essential local and international travel, etc.). Responses are ranged from 1 ("strongly disagree") to 5 ("strongly agree"). In the present sample, the translated Russian version of the CPBI showed good reliability (Cronbach's $\alpha = 0.861$).

We also used statistics on excess mortality in Russia's regions based on the average values for October, November, and December 2021 [26].

### Data analysis

The data were analyzed using RStudio and visualized using Python and RStudio. Means and standard deviations for continuous variables and frequencies and percentages for categorical variables were calculated to summarize participant characteristics. Percentages and the adjusted odds ratio with 95% CI were calculated to examine the prevalence of somatic burden. Latent profile analysis (LPA) was performed to identify the optimal number of participant

latent profiles based on eight somatic symptoms measured by the SSS-8. The optimal model was selected based on theoretical support and a combination of statistical indices, including the Akaike information criterion (AIC) [27], Bayesian information criterion (BIC) [28], and bootstrap likelihood ratio test (BLRT) [29]. For AIC and BIC, lower values show better model fit. The information criteria and BLRT were used to decide whether a model with k profiles was superior to the another less parsimonious model with k-1 profile. Entropy values were also identified. These values can range from 0 to 1, with higher values showing greater accuracy of classification and values above 0.80 signifying classification with minimal uncertainty [30]. After selecting the final latent profile solution, we examined multinomial regression models to predict compliance patterns as associated factors of somatic burden.

### Ethical considerations

This study was approved by the Ethics Committee of the School of Psychology, HSE University (minutes of the meeting of October 25, 2021). The participants gave written informed consent.

## Results

### Descriptive statistics

Participant and descriptive characteristics are presented in Table 1. The study sample comprised 10,205 persons from 33 regions of Russia (see Fig 1).

### Prevalence of somatic burden

Over one-third (37%) of the participants reported being somatised. A total of 16.8% of the respondents were free from somatic burden, 23.9% had low somatic burden, 22.2% had medium somatic burden, 17.4% had high somatic burden, and 19.7% had very high somatic burden. Fig 1 shows prevalence of somatic burden by Russia's regions, illustrating important variations ranging from 21.6% in the Primorye Territory to 49.7% in the Ulyanovsk Region.

### Latent profiles of somatic burden

Table 2 presents fit indices of the models with an increasing number of somatic burden profiles. The models with one latent profile and two latent profiles had the greatest AIC and BIC, suggesting that these models fit the data the worst. The model with three latent profiles had lower AIC and BIC values, and further separation of four or five latent profiles did not improve these fit statistics essentially. Entropy values close to 1 are preferred, evidencing better profile separation [30]. The models showed a decrease in values from one to five profiles. Although

**Table 1. Participant and descriptive characteristics.**

| Characteristic | Mean (SD) or *n* (%) |
|---|---|
| Gender, female participants, *n* (%) | 7,766 (77.5) |
| Age (in years), mean (SD) | 36.10 (14.11) |
| Educational background, university, *n* (%) | 7,443 (73.5) |
| Partnership status, being in a partnership, *n* (%) | 6,063 (59.9) |
| History of COVID-19 disease *n* (%) | 6,102 (60.1) |
| Vaccination against SARS-CoV-2, *n* (%) | 5,597 (55.2) |
| Self-rated health, mean (SD) | 3.78 (0.81) |
| Fear of COVID-19, mean (SD) | 15.97 (5.31) |
| Preventive behavior during the COVID-19, mean (SD) | 33.52 (8.44) |

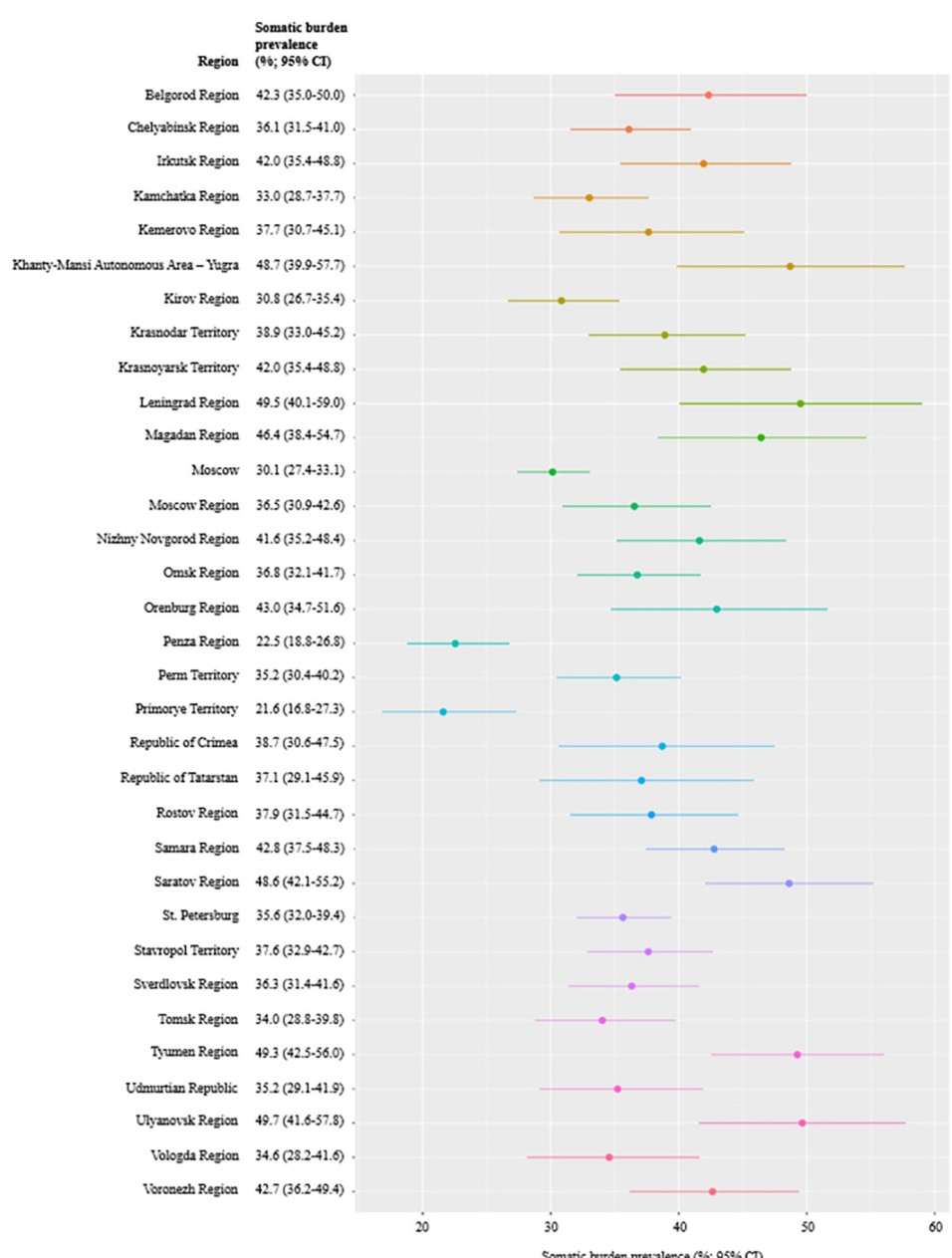

**Fig 1. Somatic burden prevalence by Russia's regions.**

**Table 2. Summary of fit statistics for latent profile models based on eight somatic symptoms.**

| Model | AIC | BIC | Entropy | BLRT |
|---|---|---|---|---|
| 1 latent profile | 232,661.81 | 232,777.57 | 1.00 | |
| 2 latent profiles | 213,661.72 | 213,842.59 | 0.84 | 0.01 |
| 3 latent profiles | 209,118.91 | 209,364.89 | 0.80 | 0.01 |
| 4 latent profiles | 207,598.86 | 207,909.95 | 0.78 | 0.01 |
| 5 latent profiles | 207,616.26 | 207,992.47 | 0.68 | 0.01 |

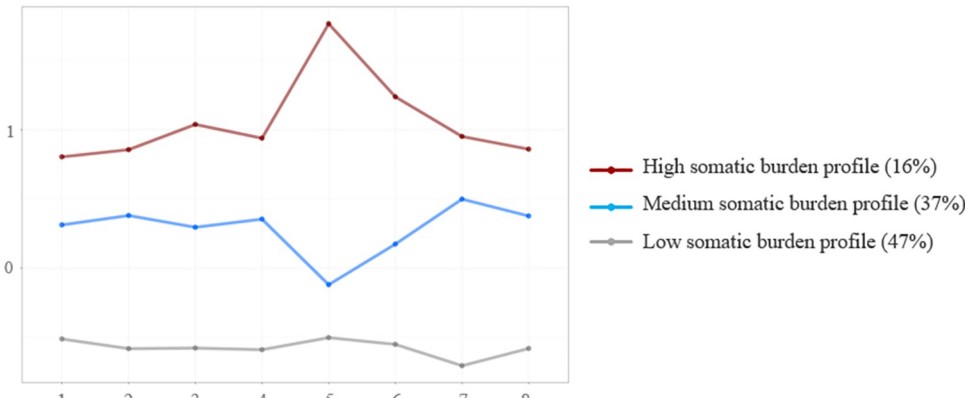

**Fig 2. The three somatic burden latent profiles characterized by their patterns of the eight somatic symptoms.**
Symptoms: 1 = stomach or bowel problems; 2 = back pain; 3 = pain in arms, legs, or joints; 4 = headaches; 5 = chest pain or shortness of breath; 6 = dizziness; 7 = feeling tired or having low energy; 8 = trouble sleeping.

the goodness-of-fit indices were not in full agreement to support a single model, we opted for the three-profile solution based on previous theoretical, empirical, and practical considerations on the separation of low, medium, and high somatisers in clinical and general population [15, 31, 32].

The response patterns characterising the three latent profiles of somatic burden are shown in Fig 2. The first latent profile includes 16% of the participants with generally high severity of stomach or bowel problems, back pain, pain in arms, legs, or joints, headaches, feeling tired or having low energy, and trouble sleeping combined with even more pronounced dizziness and extremely high levels of chest pain or shortness of breath. This latent profile can be described as a "high somatic burden profile". The second latent profile contains 37% of the participants showing moderate complaints of stomach or bowel problems, back pain, pain in arms, legs, or joints, headaches, feeling tired or having low energy, and trouble sleeping with somewhat lower mentions of dizziness and chest pain or shortness of breath. This latent profile can be referred to as the "medium somatic burden profile". The las and most numerous latent profile contains 47% of the participants and can be labelled as "low somatic burden profile", as participants in this profile reported low severity of all somatic symptoms. This suggests that for almost half of the survey sample, somatic burden might not be a typical way of responding to the COVID-19 pandemic. In Fig 2, the y-axis shows the z-scores reflecting the levels of somatic burden, while the x-axis represents eight specific somatic symptoms used for the LPA. The three lines illustrate patterns for the three somatic burden latent profiles.

## Associated factors of somatic burden

The associations of somatic burden with other variables are displayed in Fig 3. Compared with high somatisers, there was clear evidence that medium somatisers have higher education (OR = 1.21, 95% CI 1.04 to 1.40), more rarely reported history of COVID-19 disease (OR = 0.58, 95 CI 0.50 to 0.67), greater self-rated health (OR = 2.68, 95% CI 2.43 to 2.96), lower fear of the pandemic (OR = 0.96, 95% CI = 0.95 to 0.97), and live in regions with lower excess mortality (OR = 1.00, 95% CI 1.00 to 1.00). Similar, but more extensive associations were found with a larger gap in somatic burden between participants. Compared with high somatisers, low somatisers were more often male respondents (OR = 0.47, 95% CI 0.39 to 0.58), more likely to have higher education (OR = 1.29, 95% CI 1.08 to 1.54), less likely to report history of COVID-19 disease (OR = 0.46, 95% CI = 0.39 to 0.55), more frequently

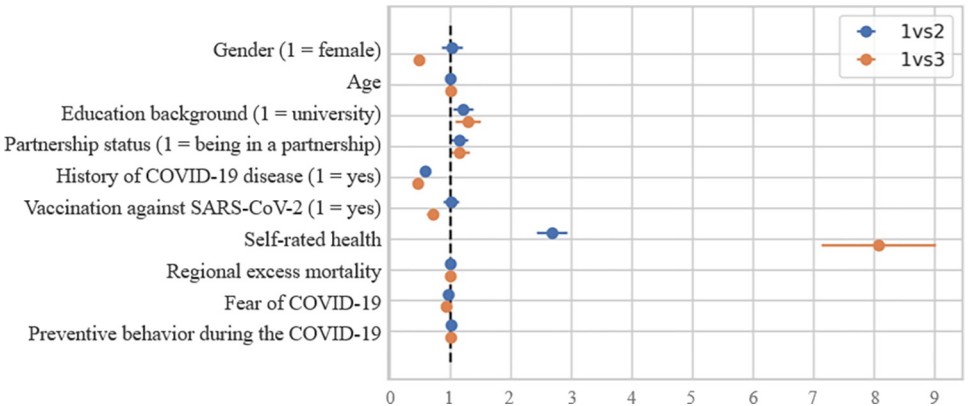

**Fig 3. Results of multinomial regression model of somatic burden latent profile on participant characteristics.**
Profiles: 1 = high somatic burden profile; 2 = medium somatic burden profile; 3 = low somatic burden profile.

reported vaccination against SARS-CoV-2 infection (OR = 0.71, 95% CI 0.61 to 0.83), greater self-rated health (OR = 8.07, 95% CI 7.13 to 9.18), lower fear of the pandemic (OR = 0.93, 95% 0.91 to 0.95), and live in regions with lower excess mortality (OR = 1.00, 95% CI 1.00 to 1.00). Thus, female gender, lower education, history of COVID-19 disease, refusing vaccination against SARS-CoV-2 infection, poorer self-rated health, greater fear of COVID-19 pandemic, and living in regions with higher excess mortality were associated with greater somatic burden.

## Discussion

To our knowledge, this study is the largest to specifically focus on examining the prevalence of somatic burden in Russia. We found that 37% of the Russians reported being somatised, with an important variation of prevalence across regions ranging from 21.6% in the Primorye Territory to 49.7% in the Ulyanovsk Region. These data show that the somatic burden in Russia during the COVID-19 pandemic is lighter than in China [6] and Brazil [7], but heavier than in Spain [3], Germany [4], and Iran [5]. A World Health Organization study evidenced that some somatic symptoms are the most common complaints in different countries, but there are culture- or geographic area-specific symptoms, such as "numbness" and "feelings of heat" in Africa, "burning hands and feet" in India, and "fatigue" in Western countries [20]. The possible differences in symptom prevalence between Russia's regions can be related to a large number of factors, such as geographic features, socioeconomic status, rates of morbidity and mortality from SARS-CoV-2 infection, residents' attitudes toward the COVID-19 pandemic, and trust in local government authorities.

We found three latent profiles describing Russians with low, medium, and high somatic burden during the COVID-19 pandemic, which corresponds to a previously discovered classification of participants with somatic complaints [15]. Surprisingly, the medium and high somatic burden profiles form a diamond-shaped pattern with dizziness and chest pain or shortness of breath increased in the highly somatising group and decreased in the moderately somatising group. There could be two possible interpretations of these findings. Dizziness and chest pain are considered as physiological symptoms of panic attacks, and panic attacks are a common phenomenon among persons with somatoform disorders [33]. This means that persons with high somatic burden can react panically to physical symptoms resembling COVID-19 disease. Dizziness and chest pain are also considered as long-term effects of SARS-CoV-2 infection [34]. Because persistent somatic symptoms are seen in patients suffering from post-

acute long COVID [35], and most respondents noted a history of the disease, some of them may have experienced post-acute COVID-19 syndrome with dizziness, chest pain, and shortness of breath.

The associated factors of greater somatic burden were female gender, lower education, history of COVID-19 disease, refusing vaccination against SARS-CoV-2 infection, poorer self-rated health, greater fear of COVID-19 pandemic, and living in regions with higher excess mortality. Previous research showed the associations of somatic burden with female gender [36] and fear of the pandemic [37], but not with lower education [38] and refusing vaccination against SARS-CoV-2 infection [39]. The fact that vaccination against SARS-CoV-2 infection, but not lower education and preventive behavior during the pandemic was associated with lower somatic symptoms evidenced the trust in medicine and good awareness about the pandemic among Russians. Importantly, the strongest associated factor was poor self-rated health, which may suggest that a person's doubts about his or her health play a key role in the formation and aggravation of somatic symptoms during the COVID-19 pandemic. The pre-pandemic study found that non-specific health complaints and a poor self-rated health in pre-adolescents were associated with greater past and future use of general practitioners [31].

Despite strengths, this study has a number of limitations. The online survey setting limits our ability to investigate the possible causes of reported somatic symptoms, which could be associated with mental or physical disorders. This study also relies on self-report data, making it impossible to accurately determine certain participant characteristics, such as their demographic features, history of COVID-19 disease, and experience of vaccination against SARS-CoV-2 infection. We suppose that many participants may have experienced the COVID-19 disease asymptomatically or, conversely, may have interpreted some somatic complaints as symptoms of the COVID-19 disease. Similarly, unvaccinated persons may misrepresent their vaccination status because of stigmatisation [40]. Future studies could rely on objective data (medical records, laboratory tests, physician evaluations, etc.). This study covered only eight somatic symptoms for which we identified latent somatic burden profiles. Previous studies included 9 [16], 12 [15], and 31 somatic symptoms [17], resulting in a greater number of somatic burden latent profiles.

## Conclusions

This study contributes to our knowledge of the prevalence, latent profiles, and associated factors of somatic burden during the COVID-19 pandemic and may extend the previous findings and contribute to improving the existing practices in psychosomatic medicine. This is also the first large-scale study of somatic symptoms among Russians, and we hope that our findings will open up prospects for new investigations of mental and physical health issues in Russia.

## Acknowledgments

We are grateful to the participants, as well as thankful to the editors and anonymous reviewers.

## Author Contributions

**Conceptualization:** Alena Zolotareva.

**Data curation:** Alena Zolotareva.

**Investigation:** Anna Khegay, Elena Voevodina, Nina Nizovskih, Vsevolod Konstantinov, Arina Malenova, Irina Belasheva, Natalia Khodyreva, Vladimir Preobrazhensky, Kristina Azanova, Lilia Sarapultseva, Almira Galimova , Inna Atamanova, Anastasia Kulik, Yulia

Neyaskina, Maksim Lapshin, Marina Mamonova, Ruslan Kadyrov, Ekaterina Volkova, Viktoria Drachkova, Andrey Seryy, Natalia Kosheleva.

**Methodology:** Alena Zolotareva, Evgeny Osin.

**Project administration:** Alena Zolotareva, Anna Khegay, Elena Voevodina.

**Software:** Igor Kritsky, Roman Ibragimov.

**Supervision:** Evgeny Osin.

**Visualization:** Igor Kritsky, Roman Ibragimov.

**Writing – original draft:** Alena Zolotareva.

**Writing – review & editing:** Inna Atamanova, Evgeny Osin.

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
