## [Decision Letter · Decision Letter 0]

17 Oct 2022

PONE-D-22-18719Somatic burden in Russia during the COVID-19 pandemicPLOS ONE

Dear Dr. Zolotareva,

Thank you for submitting your manuscript to PLOS ONE. After careful consideration, we feel that it has merit but does not fully meet PLOS ONE’s publication criteria as it currently stands. Therefore, we invite you to submit a revised version of the manuscript that addresses the points raised during the review process.

We look forward to receiving your revised manuscript.

Kind regards,

Ahmet Murt

Academic Editor

PLOS ONE

Journal Requirements:

Reviewers' comments:

Reviewer's Responses to Questions

**Comments to the Author**

1. Is the manuscript technically sound, and do the data support the conclusions?

Reviewer #1: Yes

Reviewer #2: Yes

2. Has the statistical analysis been performed appropriately and rigorously? 

Reviewer #1: Yes

Reviewer #2: Yes

3. Have the authors made all data underlying the findings in their manuscript fully available?

Reviewer #1: Yes

Reviewer #2: Yes

4. Is the manuscript presented in an intelligible fashion and written in standard English?

Reviewer #1: Yes

Reviewer #2: Yes

5. Review Comments to the Author

Reviewer #1: I think it is a good article evaluating the somatic burden.

Minor Points

- In Table 1, I think that it is not necessary to give all the cities where the participants in the study live. The table seems too long.

- Table 1 presents data on the results of the Fear of COVID-19 Scale and the СOVID-19 Preventive Behavior Index. Therefore, it would be more appropriate to give Table 1 after the explanations of these scales.

-Typos should be corrected.

Reviewer #2: In this article, authors have researched the somatic burden of Russian population during COVID-19 pandemic. The scales and number of participants are acceptable.

However, the paper needs some modifications before it can proceed to the publication stage:

1- Why didn't the authors exclude any patients with any previous psychological or psychiatric illnesses?

2- Do we have any information about previous health problems of the participants? There might appear a difference between those who have and those who don't have.

3- In somatic symptom scale, authors state that there are 4 levels of symptoms: minimal, low, medium, high, very high. However in latent profiles there are only low, medium and high levels. Can the authors explain why they reduced the number of profile classes?( (It's stated that goodness for fit model does not indicate the superiority of any other)

4- Although females are given and discussed as higher somatizers, it is said in associated factors of somatic burden that 'low somatisers were more often females'. This should be corrected.

5- Authors should explain their selection criteria for the multinomial regression model.

6- May authors also discuss the reasons for differences between different cities or regions? Do any of these regions have any specific characteristics?

6. PLOS authors have the option to publish the peer review history of their article (what does this mean?). If published, this will include your full peer review and any attached files.

Reviewer #1: No

Reviewer #2: No

---

## [Author Response · Author response to Decision Letter 0]

4 Nov 2022

Dear Reviewers,

Thank you for your help in improving our manuscript. We have made the modifications following your suggestions:

In Table 1, I think that it is not necessary to give all the cities where the participants in the study live. The table seems too long.

Thank you, for brevity we have excluded the list of cities from Table 1.

Table 1 presents data on the results of the Fear of COVID-19 Scale and the СOVID-19 Preventive Behavior Index. Therefore, it would be more appropriate to give Table 1 after the explanations of these scales.

Thank you for these suggestions, we moved Table 1 to the Results section, so that readers can see the descriptive statistics for these instruments after they read their descriptions in Methods.

Typos should be corrected.

Thank you for this suggestion. We have proofread the manuscript and had the text proofread by a native English-speaking editor who used British spelling.

Why didn't the authors exclude any patients with any previous psychological or psychiatric illnesses?

Indeed, we did not specifically exclude patients with any prior psychological or psychiatric illnesses for several reasons. Firstly, The Somatic Symptom Scale-8 used in our study fairly accurately assesses the DSM-5 somatic symptom disorder (AUC = 0,71; 95% CI = 0,66-0,77, according to Toussaint et al., 2020). This implies that more than a third of our sample (37% of participants) meet the criteria for somatic symptom disorder. Moreover, a meta-analytic review has shown that at least a third of patients with somatoform disorders have comorbid anxiety and depressive disorders (Henningsen et al., 2003). Given this, somatic symptoms and mental disorders emerge as quite difficult to differentiate, and we believe that doing it would make the sample of individuals with somatic symptoms less representative.

Secondly, our study was carried out in an online setting, where we could not be sure that the participants would objectively assess their psychological state or honestly report a psychiatric diagnosis. And, finally, there is evidence indicating that many disorders are diagnosed in Russia extremely rarely: for instance, only 1-4% of cases of bipolar affective disorder, depression, anxiety disorders are diagnosed by specialists (Martynikhin, 2021): as a result, most Russian participants with mental comorbidities would probably never be aware of the fact.

However, we totally agree with you that excluding patients with a history of psychiatric illnesses could affect our findings and we described this as a limitation of the current study (“The online survey limits our ability to highlight possible physiological causes of reported somatic symptoms, which may be associated with mental and physical disorders”) and a prospect for future research (“Future studies should rely on objective characteristics (medical records, laboratory tests, physician evaluations, etc.)”). 

Henningsen P., Zimmermann T., Sattel H. (2003). Medically unexplained physical symptoms, anxiety, and depression: a meta-analytic review. Psychosomatic Medicine, 65, 528–533. https://doi.org/10.1097/01.psy.0000075977.90337.e7

Martynikhin, I. A. (2021). The use of ICD-10 for diagnosing mental disorders in Russia, according to national statistics and a survey of psychiatrists’ experience. Consortium Psychiatricum, 2(2), 35-44. https://consortium-psy.com/jour/article/view/69/pdf

Toussaint A., Hüsing P., Kohlmann S., & Löwe B. (2020). Detecting DSM-5 somatic symptom disorder: criterion validity of the Patient Health Questionnaire-15 (PHQ-15) and the Somatic Symptom Scale-8 (SSS-8) in combination with the Somatic Symptom Disorder – B Criteria Scale (SSD-12). Psychological Medicine 50, 324–333. https://doi.org/10.1017/S003329171900014X

Do we have any information about previous health problems of the participants? There might appear a difference between those who have and those who don't have.

Our questionnaire contained only one question about the respondents' objective health status ("Do you think you were sick with COVID-19?"). We did not ask about chronic illnesses and other possible medical causes of somatic symptoms for the same reasons as we did not ask about psychiatric illnesses. We agree that this is a very serious question and we also noted this in the Limitations section: “”

In somatic symptom scale, authors state that there are 4 levels of symptoms: minimal, low, medium, high, very high. However in latent profiles there are only low, medium and high levels. Can the authors explain why they reduced the number of profile classes? (It's stated that goodness for fit model does not indicate the superiority of any other).

The authors of the Somatic Symptom Scale-8 distinguish five levels of somatic burden: minimal, low, medium, high, and very high (Gierk et al., 2013). In choosing the number of latent profiles, we relied on a combination of information criteria, entropy, and interpretability of the resulting classes. In particular, the 5-profile model had higher AIC and BIC values, suggesting that this number of profiles is excessive. Based on a combination of theoretical and empirical considerations, in line with some existing studies (Burri et al., 2017; Eliasen et al., 2018; Kato et al., 2010), we opted for a 3-profile model that, as we believe, is parsimonious and at the same time sufficiently theoretically clear.

Burri A, Hilpert P, McNair P, Williams FM. Exploring symptoms of somatization in chronic widespread pain: Latent class analysis and the role of personality. J. Pain Res. 2017; 10:1733–1740. https://doi.org/10.2147/JPR.S139700

Eliasen M, Schröder A, Fink P, Kreiner S, Dantoft TM, Poulsen CH, et al. A step towards a new delimitation of functional somatic syndromes: A latent class analysis of symptoms in a population-based cohort study. J. Psychosom. Res. 2018; 108:102–117. https://doi.org/10.1016/j.jpsychores.2018.03.002

Gierk B., Kohlmann S., Kroenke K., Spangenberg L., Zenger M., Brähler E., et al. The somatic symptom scale-8 (SSS-8): A brief measure of somatic symptom burden. JAMA Internal Meicine, 174(3), 399–407. https://doi.org/10.1001/jamainternmed.2013.12179

Kato K, Sullivan PF, Pedersen NL. Latent class analysis of functional somatic symptoms in a population-based sample of twins. J. Psychosom. Res. 2010; 68(5):447–453. https://doi.org/10.1016/j.jpsychores.2010.01.010

Although females are given and discussed as higher somatizers, it is said in associated factors of somatic burden that 'low somatisers were more often females'. This should be corrected.

Thank you so much for helping us see and correct this mistake, which was a typo. While proofreading the text, we found another similar mistake in the description of the latent profiles (‘more frequently reported history of COVID-19 disease’ instead of ‘more rarely reported history of COVID-19 disease’). All errors have been corrected. 

Authors should explain their selection criteria for the multinomial regression model.

We tried to explain selection criteria for the multinomial regression model by adding the following sentence to the Introduction: ‘Specifically, we presumed that somatic burden during the COVID-19 pandemic is associated with known pre-pandemic risk factors (female gender, older age, not being in a partnership, lower education background) [18; 19; 20], but is more dependent on health- and pandemic-related risk factors (poorer self-rated health, history of COVID-19 disease, refusing vaccination against SARS-CoV-2 infection, greater fear of COVID-19 pandemic, living in regions with higher excess mortality, and preventive behavior during the COVID-19 pandemic)’. We also cited additional sources to justify the choice of predictors.

May authors also discuss the reasons for differences between different cities or regions? Do any of these regions have any specific characteristics?

We think there could be many reasons for differences between different cities or regions, as we write about in the Discussion: ‘Variation in prevalence between Russia’s regions can be related to a large number of factors such as geographic features, socioeconomic status, rates of morbidity and mortality from SARS-CoV-2 infection, residents’ attitudes toward the COVID-19 pandemic, and trust in local government authorities’. This is a very important and interesting question, which needs to be explored in the following studies, as the procedures and methods of diagnosing the COVID-19 disease, the patient routing protocols in case of positive results, and the algorithms for inpatient and outpatient treatment were the same throughout Russia (in accordance with the orders of the Ministry of Health).

Once again, we thank you for your help and we hope that we have corrected all the inconsistencies and explained all the shortcomings.

With best regards,

authors.

---

## [Decision Letter · Decision Letter 1]

12 Dec 2022

PONE-D-22-18719R1Somatic burden in Russia during the COVID-19 pandemicPLOS ONE

Dear Dr. Zolotareva,

Thank you for submitting your manuscript to PLOS ONE. After careful consideration, we feel that it has merit but does not fully meet PLOS ONE’s publication criteria as it currently stands. Therefore, we invite you to submit a revised version of the manuscript that addresses the points raised during the review process.

We look forward to receiving your revised manuscript.

Kind regards,

Ahmet Murt

Academic Editor

PLOS ONE

Journal Requirements:

Additional Editor Comments:

Dear Authors,

Our reviewers stated that the link for your data was not active. Can you please check and assure that your data is available as you stated in your manuscript.

Thank you for your contribution to our journal.

Reviewers' comments:

Reviewer's Responses to Questions

**Comments to the Author**

1. If the authors have adequately addressed your comments raised in a previous round of review and you feel that this manuscript is now acceptable for publication, you may indicate that here to bypass the “Comments to the Author” section, enter your conflict of interest statement in the “Confidential to Editor” section, and submit your "Accept" recommendation.

Reviewer #1: All comments have been addressed

Reviewer #2: All comments have been addressed

2. Is the manuscript technically sound, and do the data support the conclusions?

Reviewer #1: (No Response)

Reviewer #2: Yes

3. Has the statistical analysis been performed appropriately and rigorously? 

Reviewer #1: (No Response)

Reviewer #2: Yes

4. Have the authors made all data underlying the findings in their manuscript fully available?

Reviewer #1: (No Response)

Reviewer #2: No

5. Is the manuscript presented in an intelligible fashion and written in standard English?

Reviewer #1: (No Response)

Reviewer #2: Yes

6. Review Comments to the Author

Reviewer #1: (No Response)

Reviewer #2: The suggestions in the previous round were all applied. The authors should not forget to make the data of this study fully available.

7. PLOS authors have the option to publish the peer review history of their article (what does this mean?). If published, this will include your full peer review and any attached files.

Reviewer #1: No

Reviewer #2: No

---

## [Author Response · Author response to Decision Letter 1]

28 Jan 2023

Dear Reviewers,

Thank you for your help in improving our manuscript!

We had probably included a peer-review link only. We have now replaced it with a link to our dataset in the OSF repository (https://osf.io/9hw52/) that is publicly available without restrictions.

Best regards,

the authors.

---

## [Editor Report · Decision Letter 2]

14 Feb 2023

Somatic burden in Russia during the COVID-19 pandemic

PONE-D-22-18719R2

Dear Dr. Zolotareva,

We’re pleased to inform you that your manuscript has been judged scientifically suitable for publication and will be formally accepted for publication once it meets all outstanding technical requirements.

Kind regards,

Ahmet Murt

Academic Editor

PLOS ONE

Additional Editor Comments (optional):

Thank you for your efforts to answer the reviewers' concerns.
---

## [Editor Report · Acceptance letter]

28 Feb 2023

PONE-D-22-18719R2 

Somatic burden in Russia during the COVID-19 pandemic 

Dear Dr. Zolotareva:

I'm pleased to inform you that your manuscript has been deemed suitable for publication in PLOS ONE. Congratulations! Your manuscript is now with our production department. 

Kind regards, 

on behalf of

Dr. Ahmet Murt 

Academic Editor

PLOS ONE